# "If We Don't Listen to Them, We Make Them Lose More than Money:" Exploring Reasons for Underreporting and the Needs of Older Scam Victims

Katalin Parti *[ID] and Faika Tahir [ID]

Department of Sociology, College of Liberal Arts and Human Sciences, Virginia Tech, Blacksburg, VA 24061, USA
* Correspondence: kparti@vt.edu

**Abstract:** Highly manipulative online and telephone scams committed by strangers target everyone, but older individuals are especially susceptible to being victimized. This study aimed to (1) identify why older individuals decide not to report scams and, in parallel, (2) explore the needs of victims. Thirty-five interviews were conducted with Virginia residents who were 60 years or older in 2021. The interpretive phenomenological analysis of the semi-structured interviews revealed that victims are reluctant to report crimes or ask for help from their family or community because much-needed emotional, educational, and technical help is often inaccessible or inadequate. In particular, we found that family responses tend to intrude on privacy, community responses are not meaningful or are non-existent, police responses are inadequate, and prevention programs are inaccessible and not specified to meet the needs of older age groups. We recommend developing age-appropriate prevention and education programs, by applying the intergenerational group approach, and actively listening to victims' concerns before deciding what means of help should be applied. Research implications and recommendations are presented.

**Keywords:** scam; fraud; older individuals; reporting; interpretive phenomenological analysis; needs; victim; community

## 1. Introduction

In May 2022, former FBI director William H. Webster warned of scams targeting older individuals (Federal Bureau of Investigation 2022). "If it can happen to me, it can happen to you," said Webster, who refused to pay $50,000 to the scammer before getting the car and multimillion dollars he allegedly won in the lottery. Instead, Webster worked with the FBI to catch the scammer, who is now serving a prison sentence (ibid.). This story ends here, but each year there are millions of older Americans who are targeted by various types of scams, such as tech support fraud, confidence/romance scams, lottery/sweepstakes/inheritance scams, government impersonation, investment scams, and cryptocurrency fraud (Federal Bureau of Investigation 2021), without experiencing such a lucky ending.

Scams are technology-facilitated frauds when the offenders, who are strangers to the victims, intentionally deceive victims by misrepresenting, concealing, or omitting facts about promised goods; services; or other physical, mental, or emotional expectations that are nonexistent, unnecessary, or deliberately distorted for monetary gain (adapted from Beals et al. 2015). Financial fraud differs from financial exploitation/abuse that is committed by caregivers or other trusted individuals (Hall et al. 2016). Scammers utilize technology, such as the internet and telephone (cell or landline), to reach out to targeted people and manipulate them into sharing personal information, such as social security, credit card, and other personal identification numbers, via email or phone (Federal Trade Commission Consumer Advice 2020). Scammers can approach their victims via social media or dating websites to gain their trust (Hanoch and Wood 2021). Some offenders pretend they are the victims' grandchildren or friends who are in trouble and convince victims to send money

to them (Payne 2020). Others pretend they are legitimate agencies, such as the Internal Revenue Service (IRS; the agency responsible for enforcing the internal revenue or tax laws in the US) or insurance companies, and tell victims that they owe money, which they can pay via gift cards (Federal Trade Commission Consumer Advice 2020).

Although scammers target everyone independently of age, their scams disproportionately affect older people. According to the FBI's Elder Fraud Report, victims aged 60 and above represented 21.5% of all fraud victimization, sustaining 30.4% of total fraud losses and resulting in approximately $1.7 M losses among older persons in 2021 (Federal Bureau of Investigation 2021). This represents an increase of approximately $720,000 in losses reported in 2021 versus 2020 (ibid.). Apart from financial loss, victims face other consequences, such as anxiety, depression, and chronic stress (Lichtenberg et al. 2013; Lichtenberg et al. 2016; DeLiema et al. 2020), which, in turn, negatively impact social relationships (Bailey et al. 2021). The vast majority of scams remain unreported (Button et al. 2014), yet, when it comes to older adults, they are reported even less often (Pak and Shadel 2011; Van Wyk and Mason 2001).

Older people are victimized by scammers relatively easily for various reasons. First, since they adopt new technologies later in life, their online interactions are usually newer compared to younger generations (Mitzner et al. 2019). Second, older adults have a 71% share of the total wealth in the US economy, meaning that they represent a lucrative generation to target and have much to lose financially (Vandenbroucke and Zhu 2017). Moreover, during COVID-19, most businesses and work shifted to remote online mediums, resulting in older adults being even more exposed to online fraud.

Several studies highlight the nature and risk factors of being scammed online (for example, DeLiema et al. 2020; Van Wilsem 2013; Wood and Lichtenberg 2017; Parti 2022); however, there are very few studies that address the causes of the underreporting of these crimes (National Statistics UK 2019). With few notable exceptions (Cross et al. 2016; Kopp et al. 2015), most studies only apply quantitative methodologies in the study of financial scams against older people, thus missing the victims' narratives and interpretations of needs as well as the reasons given for not reporting (for example, Li et al. 2022; Zhang and Ye 2022; for a systematic review see Norris et al. 2019). The present research fills this gap by providing insight into scam victims' narratives and helping us to determine why older scam victims are reluctant to report. The study aimed to explore and share victims' experiences and needs, under the assumption that a better understanding would make victims more likely to ask for assistance from trusted individuals or from the police and anti-fraud agencies. Data were gathered through in-depth, semi-structured interviews with victims and individuals in close proximity to the victims. By analyzing scam-related experiences, this paper provides firsthand information that individuals who are in close proximity to older victims—such as relatives, caretakers, and community members, as well as authorities—can utilize in their efforts to develop solutions to scams targeted against older individuals.

## 2. Theoretical Framework

Several theoretical perspectives are available in the scientific literature regarding factors that affect victims' decisions about reporting crimes. The factors that may influence this decision are classified, according to Goudriaan et al. (2006), as economic, psychological, and neighborhood or societal factors. The *economic perspective* assumes that an individual's decision to report a crime is influenced by the expected benefits and costs (Skogan 1984; Bowles et al. 2009). As a result, victims will choose to report if it is the most viable alternative to dealing with the victimization's consequences (i.e., the expected benefits minus the expected costs). Tarling and Morris (2010) suggest that victims consider how long it will take to report the crime and whether assets can be recovered. Victims who have suffered financial losses (Gutierrez and Kirk 2017) or physical injuries are also more likely to come forward (Baumer and Lauritsen 2010; Mayhew 1993). In contrast, victims who suffer non-financial losses due to trivial crimes may feel discouraged from reporting (Asiama and Zhong 2022).

According to the *sociological perspective*, patterns in crime-reporting behavior can also be explained by the social environment, including neighborhood characteristics (Goudriaan et al. 2006). The degree of urbanization (Torrente et al. 2017), social cohesion (Hart and Colavito 2011), ethnic composition (Gutierrez and Kirk 2017), and residential mobility (Schnebly 2008) all play a role in crime reporting. The reporting process is also affected by several sociodemographic factors. For example, women (Baumer and Lauritsen 2010) and victims with a partner (Tarling and Morris 2010; Goudriaan et al. 2004; Boateng 2018) are more likely to report crimes. However, the picture is more diverse when it comes to different types of crimes. For instance, Lizotte (1985) found that highly educated females are reluctant to report sexual harassment or assault to the police because they fear losing their social and economic status. The effect of age on reporting is, again, mixed. Acierno et al. (2001) and Heath et al. (2013) found that older individuals are less likely to report crimes to the police, while others (Goudriaan et al. 2006; Gutierrez and Kirk 2017; Baumer and Lauritsen 2010; Torrente et al. 2017; Goudriaan et al. 2004; Tolsma et al. 2012) have found that the opposite is true. Some researchers (Asiama and Zhong 2022) suggest that older people report crimes more often, as they tend to lose more money and, thus, seek to regain such assets through reporting to the police. Others (Van de Weijer et al. 2019) have found that victims are significantly less likely to report consumer fraud to the police when they are older, single, students, or bisexual rather than younger, married, employed, and heterosexual, respectively.

As part of the *psychological perspective*, friends, family members, and colleagues can significantly influence a victim's social environment. As a result of psychological factors involved in the decision-making process, victims are not always able to make rational decisions about whether to report. Stress and fear may prevent victims from rationally calculating the cost–benefit ratio (Goudriaan et al. 2006). The chances of victims reporting a crime might be reduced if they feel guilty about their role in the crime; for example, if they believe their behavior may have precipitated it (Cross 2015). It is also possible for victims to feel ashamed, stigmatized, and embarrassed that they were victimized, which makes it difficult for them to report to the police or seek assistance (Cross et al. 2016; DeLiema et al. 2017). A challenging and prolonged criminal justice process may also discourage reporting (Boateng 2015). Furthermore, online crimes involving offenders and victims with no prior relationship with one another are less likely to be reported to the police than traditional crimes (Van de Weijer et al. 2019).

Cybercrime is one of the types of crime that is the least likely to be reported. Using a cross-sectional sample of the Dutch population, Van de Weijer et al. (2019) discovered that 37.5% of all crimes are reported to the police, whereas only 22.7% of all cybercrimes are reported, and that such reporting is primarily to organizations *other* than the police. Most victims of identity theft report their victimization to organizations other than the police (82.3%). However, consumer fraud and hacking victims rarely report their losses to the police or other organizations. The low prevalence of reporting cybercrime victimization is consistent with previous studies (Domenie et al. 2013). Victims in these situations do not wish to endure the stress of the criminal justice process or waste time with that since there is a minimal chance of even identifying the offender (Asiama and Zhong 2022; Goudriaan and Nieuwbeerta 2007).

There has also been some evidence that victims' attitudes toward the police influence reporting (Goudriaan et al. 2004; Van de Weijer et al. 2019; Leitgöb-Guzy and Hirtenlehner 2015; Kerley and Copes 2002). When victims have a positive attitude toward the police or confidence in them, they are more likely to report their victimization. Conversely, victims who are dissatisfied with how the police have handled a previous report are less likely to contact law enforcement (see Domenie et al. 2013). For example, individuals who have been victimized more than once are less likely to contact the police, as the police may not have responded satisfactorily to previous situations (Asiama and Zhong 2022). In addition, they may assume that the police are ineffective and inefficient (Langton et al. 2012), or they do not want to be seen as "troublemakers" (Weisel 2005).

When police and other agencies are not the primary sources of help, interpersonal relationships can provide protection and help prevent and mitigate the consequences of scams. Research suggests, however, that asking trusted individuals for assistance can be just as uncommon as reporting to the police (DeLiema et al. 2020). For instance, the results of DeLiema and colleagues' (ibid.) randomly selected national sample of Americans aged 50 years and older found that Hispanics are less likely to report fraud victimization than other racial/ethnic groups. Although social integration was not a significant protective factor in this study (ibid.), a deeper-than-average embeddedness in family networks for Hispanic respondents may help protect them from fraud attempts. However, research is not conclusive in corroborating this finding and further investigation of the matter is recommended.

In the following section, we describe the sample and methods of the current qualitative research and discuss the results in light of previous studies. As seen above, several quantitative studies using survey methods have examined victim profiles and risk, as well as protective factors against scam victimization among older individuals (e.g., DeLiema et al. 2020; James et al. 2014; Holtfreter et al. 2014). However, fewer studies have investigated underlying factors from the victims' point of view or examined the reasons for (a lack of) reporting victimization or seeking help in general (Cross et al. 2016; Kopp et al. 2015). This paper aims to bridge this gap by exploring the needs of older scam victims among a relatively small sample of Virginia residents. Older individuals were interviewed about their victimization, and their needs concerning reporting or asking for assistance were investigated. We hope to expand the current understanding of the barriers to reporting among this group and to equip individuals and agencies with more tools for assisting older scam victims.

## 3. Materials and Methods

The current research focused on scams targeting older adults and questions related to their lived experiences as victims of these crimes. The participants described the phenomenon from the perspective of the primary informant (if they were targeted/victimized) or the secondary informant (if they were family, cohabitants, neighbors, or friends of the victims). The understanding of the phenomenon is presented through the researchers' best estimation of the experiences. The interpretive phenomenological approach utilized, as described by Smith et al. (2009); see also Smith (1996), provides the liberty to add both participants' and researchers' perceptions and understanding of the events, as well as an empathetic approach to describing them. In addition, the researchers sought to understand why older victims are reluctant to report or ask for help from the shared perspective of primary and secondary informants; that is, both groups provided narratives and interpretations of the same phenomenon and shared experience. Interpretive phenomenological analysis helps to explain the underreporting of scams by factoring in the victims' perspectives and the meaning they attach to the experiences. It focuses on the lived experiences of the respondents and how their scam victimization shapes a shift in attitude within the person or their environment.

### Sampling Strategy

The target population for this research was adults aged 60 and above who had been targeted by scams (primary informants), or people who knew someone who had been targeted by scams (secondary informants). Primary and secondary informant knowledge may overlap in the interviews, as many participants who experienced scams firsthand were also closely related to individuals who had been similarly targeted.

Three strategies were adopted for recruiting participants for an in-depth interview: a volunteer sampling method, opportunistic or emergent sampling, and snowball sampling. As a first step in the sampling process, participants voluntarily agreed to participate in an in-depth interview after taking a short initial survey. An online link to the survey was sent to the Virginia Chapter of the American Association of Retired Persons (AARP) and

to local senior living facilities in New River Valley, Virginia, where it was circulated to senior home residents. The survey asked about basic demographic information, as well as whether the respondent had been targeted or victimized by scams. Finally, the survey asked whether the participants wished to share their experiences with the researchers in a follow-up interview. In the latter case, there was a space to add a preferred way of communication (email address or telephone number). In the next sampling phase (opportunistic or emergent sampling), the researchers arranged stalls at local farmers' markets and other grocery store locations. To attract attention, we used banners for the research and research affiliations and a board stating, "Are you sixty or above?". This allowed us to attract participants and discuss the research aims and objectives. After several visits to farmers' markets, researchers recruited 21 participants; however, only 13 were interviewed in the end. An additional 11 participants were recruited from local independent and assisted living facilities, out of whom four agreed to be interviewed. A further 11 participants were recruited through participant referrals or snowball sampling. Snowball sampling was the most effective method for contacting potential participants who had stopped using the internet after experiencing scams and were not otherwise reachable. After victimizations, victims often lack trust in online and phone communication with strangers, which makes it difficult to access this population directly. Hence, to reach additional potential interviewees, we asked interviewed participants to establish connections between the researcher and further interview participants. Using this method, the recommendation from a trusted friend or acquaintance bolstered further potential interviewees' trust in participating in the interview. The justification for different sample strategies is, as Patton (1990) emphasizes, to obtain samples using different methods to reach a maximum number of participants and gather information-rich data for a systematic study (cited in Bailey 2007, pp. 64–65). As a result of the multiple-step sampling strategy we applied, 35 interviews were conducted in the summer of 2021. Interviews ranged from 15 to 72 min, with a 34 min mean length. Most interviewees identified as female ($n$ = 22), with the remainder being 13 male participants. One participant was of African American race, and the others were white/Caucasian. All participants were 60 years of age or above and residing in the Commonwealth of Virginia. Most participants talked about others' experiences (secondary informants, $n$ = 19), but 16 interviewees shared their own victimization through scams (four primary informants also shared others' experiences). Despite the fact that not everyone was victimized, every participant was, indeed, approached by scammers multiple times—via either phone, computer, or social media. Participants' age ranged from 60 to 85; however, they often talked about the victimization of older (85–95 years old) friends and family members.

The research was reviewed and approved in accordance with ethical guidelines by the Institutional Review Board of Virginia Tech (protocol code #21–415, 15 June 2021). Due to the COVID-19 pandemic, all interviews were conducted and recorded through the video conferencing platform Zoom. All participants were informed about their interview date and time, and were requested to set aside that time for the interview in whatever location would be convenient and without interruptions. An informed consent form was sent, via email, to participants 24 h before the interview. In addition, the form was read by the interviewer to the participant before the interview to ensure understanding. Participation was voluntary and confidential, with no compensation provided.

To conduct the interviews, a semi-structured guideline was created. Questions were formulated based on a literature review aimed at locating the research gaps. In addition, four focus group sessions were conducted with a total of 16 participants: three focus groups with caretakers at two local retirement homes and one focus group with family members of victimized individuals—all recruited via the aforementioned retirement homes. The focus groups helped set up interview questions and clarify the research gaps. After an interview guideline was created, the researchers sent the questions to a pool of local community members (who were not part of the interviews) as a pilot study. As a result, ambiguous questions were changed and made more straightforward for better understanding.

After the data collection phase, the research team reviewed the video recordings, which had to be converted into audio files for further transcription. Audio files were transcribed in two stages: through transcription software (Zoom v. 5.0.5's own transcription through closed caption and Otter.ai v. 9) and human help. The research team reviewed the transcripts for any discrepancies and compared them with the interview audio files. Due to the relatively small sample size, line-by-line coding was applied to identify index codes and for the microanalysis of codes (inductive coding). Atlas.ti v. 9 software was used for coding and categorizing codes into subgroups. The codes were reread and recoded when necessary to generate a final code. Every interview was independently coded by at least two research team members. An additional coder was applied to resolve occasional inconsistencies and achieve intercoder reliability. In a back-and-forth process, the research team reviewed and reworked code lists for consistency and reliability. Eventually, codes were divided into three large categories of emotional, educational, and technical reasons/needs responsible for the underreporting of scams (Figure 1).

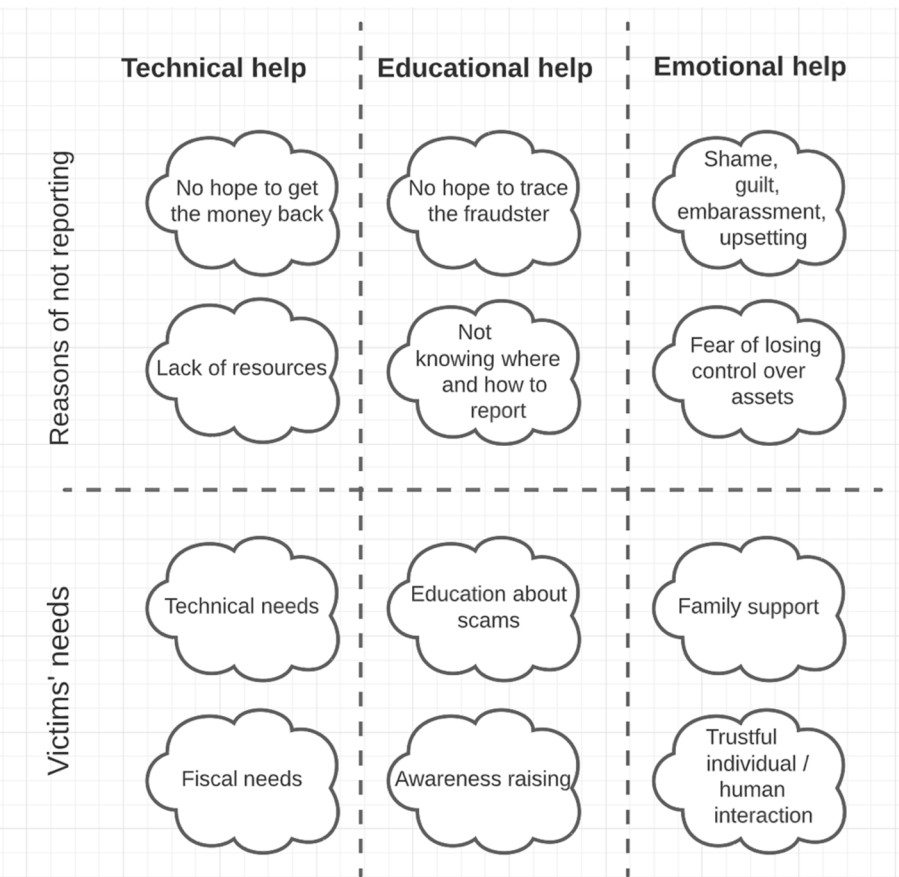

**Figure 1.** Main codes.

## 4. Results

Interview participants shared a wide range of scam activities from winnings schemes (imposters call people to notify them they won the lottery, a car, a heritage, etc.) or buying schemes (scammers want to buy something from the victim but the latter should pay them a dollar amount in advance), through company impersonation (the caller poses as a representative of a well-known company, bank, or charity agency and urges the receiver to send money or gift cards, buy goods or gifts, or pay for services offered). The latter included credit card scams ("your credit card was compromised") and IT-support scams (criminals pose as technology firm support representatives and offer to fix non-existent computer issues). Grandparent and romance scams were also mentioned, with the former being much more frequent than the latter. Other scam types included arrest order and

identity check scams (where an email communicates that the victim is going to be arrested unless they verify personal information or pay a ransom to the scammer). Interviewees provided information about successful scamming activities that cost victims $500–$40,000. Most of the time, victims did not suffer any financial loss; however, the emotional toll the scam inflicted was significant regardless of whether it was attempted or successful.

*4.1. Emotional Needs*

4.1.1. Theme 1: "I Was Scammed Because I Was Naïve"—Self-Blaming Attached to Victimization

Even if they do not lose money, most victims still need psychological support to overcome the emotional impact of scams (DeLiema 2018). This need is evident in the vocabulary participants used. Feelings of embarrassment, shame, and guilt were constantly present in the descriptions of victimization provided by both primary and secondary informants. Other feelings mentioned were anger and self-deprecating emotions, such as feeling idiotic and fooled, dumb, stupid, devastated, overwhelmed, and worthless. Falling for the scams lead to participants losing confidence in their ability to govern their own lives, judge situations, or to even make simple decisions, such as whether or not to answer emails or phone calls. Some victims felt the offender had completely taken over their lives. Other times, the severe sense of self-blame was coupled with a burdensome feeling of guilt toward family members whose inheritance has been decimated due to the victim's decision. In these situations, it is tough to share what happened, not to mention how it happened. In hindsight, victims might be able to identify the red flags; however, the fact that they missed them during the entire scam scenario leaves them reluctant to disclose their experiences. It became evident from the interviews that emotional and psychological needs overrode the rationale of reporting the scam or simply talking about it to others.

> *"The first question is, how do I get that money back? . . . And the second one is, boy, have I been fooled? I don't want to tell anybody about this."*

> *"They're ashamed to report because they were dumb enough to fall for it."*

> *"They might just go for days thinking how idiotic and how stupid and so forth they were."*

In romance scams, victims lose not only money but also the perceived love and affection of a cherished individual who turned out to be a scammer, which can be further emotionally devastating.

> *"She sent thousands of dollars to somebody that she was in love with, you know, on the internet, and she lost it all. And in that case, she's really ashamed to report; more so than if she just responded to an erroneous Amazon bid and lost money that way."*

Self-blame is a severe barrier to reporting to the police or other authorities. Aside from the embarrassment, victims might feel it was all their fault that the scam was successful, as they were the ones who decided to transfer the money.

> *"Depending on the amount of loss, I might be hurt financially, but it's all my fault."*

Participants were well aware that digital skills had to be learned later in life for older people. Some participants suggested that the self-blame for being scammed among older individuals is rooted in their inadequate technical skills or, once online, their becoming overconfident in sending or spending money, or even their willingness to talk to people online.

> *"They don't want to [report] because they're ashamed that they have learned a new skill, and then they got taken. A lot of people here are learning the computer. They've learned it, and then they go hog-wild."*

4.1.2. Theme 2: "She Has Been Reduced to a Level of a Small Child"—Family Reactions

Becoming emotionally and financially dependent due to scam victimization was a recurring theme in the interviews. The devastating nature of scam victimization comes

not only from losing significant assets but also from affecting one's financial and physical independence. It is clear from the idiomatic choices of participants, as terms such as "confession" and "admitting" that they had been "fooled" are often a sign of shame and self-blame and painful emotions when facing family members. When the "confession" meets with the family members' blame of the victim, it can be even more arduous to face the loss of financial independence, which is a protective measure that families tend to create to avoid subsequent scams.

> *"I confessed it to my children, 'Dad, who is supposed to be very proficient in finances and knows what he's doing.'–and they would bet their bottom dollar that I would not do what I did. And so that was stated."*

Losing financial independence can be an annihilating experience, especially when, according to the usual family dynamics, the older generations are assumed to be the wiser and more responsible ones. Situations became especially dire when family members (primarily descendants) decided that the only way to avoid revictimization was to confiscate bank cards or other financial assets from victims.

> *"My brother-in-law took their security cards and their credit cards. This way, they couldn't get in trouble anymore."*

> *"She's the younger sister, and she's not going to be bossed by our older one, even if they're all in their 70s."*

Participants used the metaphor of homelessness because of losing retirement funds to scammers. Unfortunately, this is not far from the reality since, according to the FBI, victims above 65 lost $18,246, on average, to scammers in 2021 (Federal Bureau of Investigation 2021); this can equate to someone's entire retirement fund and easily lead to their dependence on others. Still, even in such helpless situations, the victim does not want to ask for help because embarrassment is a more compelling feeling than the need to seek justice.

> *"And it's like homelessness, it's like not having enough money to feed your children. People don't want to claim free or reduced lunch because it's embarrassing."*

Some victims experience losing status within the family because of their remaining assets being taken over by family members. It was as if the victim became a child again, utterly dependent on others.

> *"That was devastating. Devastating. Devastating all the way around. One of the sons took her checkbook. The other son has her credit cards. So now she's completely dependent upon her two kids. 'I can't go to the grocery store. I can't do this.' You know, they're reduced to a small child who has been given an allowance."*

It is unsurprising, after all, that victims are reluctant to talk about their experiences with family members because of the fear of losing financial independence, which would be worse than losing some money to scams.

> *"If I was going in and saying, you know, 'Why did you spend this money?' or 'Why did you spend that money?' That, you know, there's a fear that you're going to try and take the control away from them."*

4.1.3. Theme 3: "Once You Get Scammed, You Become Un-Trustful and Unable to Get Help"—Trust Issues

Losing trust was a common theme in victims' nexus with their families. According to participants' evaluation, after disclosing that money was lost to scams, family members often became angry, impatient, and confused about what to do. In addition, family members often did not confirm or acknowledge the difficulties the victims had faced.

> *"My husband's response to his father being scammed was anger and impatience with his dad. Because his dad was a brilliant man. It was difficult for him to think his dad was that easily seduced."*

Trust issues also arose when people in close proximity—e.g., friends, neighbors, relatives—tried to re-evaluate the situation and intervene in pursuit of de-escalation. It was not uncommon that victims were in denial of the crime and their victimization. Hence, when people tried to intervene, they rejected the help offered, causing a loss of trust both on the victims' and the helpers' sides. In the case below, the victim was convinced that he had won a sweepstakes. He was about to hand over an "advance payment" to scammers in a mall parking lot in order to receive the prize he had "won."

> *"At that point, he still wasn't convinced that it wasn't real. I sat and watched; his wife came. And I think she was talking to him on the phone. And she couldn't really get him to believe her. So she got out of the car, went over and talked to him at the window of the car. And, you know, she really had to scream at him to get him to believe."*

In other cases, trustful relationships broke within families where the victim, already scammed many times, kept answering phone calls and sending money to scammers.

> *"His sons did talk to him quite a bit. And he said that he wasn't going to send anymore but he kept doing it."*

> *"We ran into her about six months later, downtown. [ . . . ] She was furious with us because we kept her from winning all that money. It was our fault. [ . . . ] You know, she's won the lottery and she's gonna get all this money."*

Once scammers successfully persuaded victims into making the initial payment, victims were convinced that they had done the right thing, as it was a legitimate transaction. The denial mechanism works like a vicious cycle and makes them more susceptible to being re-scammed in the future. This is the point where community members can help.

### *4.2. Educational Needs*

#### 4.2.1. Theme 4: "They Need Someone to Help Decipher What Has Happened"—Community Reactions

Community members, such as friends, neighbors, and trusted individuals, can provide valuable insight for targeted people who might need a second opinion about an ongoing situation.

> *"I think if a senior becomes unable to make a good decision, then that's when family should get involved. And the children should know when it's time they need to. I know a lot of people are not, their kids do not take it. They are not around much. And I think that may be a problem."*

One participant shared a story in which friends and neighbors joined together to successfully prevent a targeted community member from paying the offender. For such joint missions, informants emphasized the power and responsibility of the community. They asserted that the whole community needs to be educated about the nature and *modus operandi* of scams.

> *"How do I know it? Because I'd heard about it. And, you know, his neighbor and I immediately agreed, once he drove off, we're like, 'No [laughs], we are not going to let him do this!'*

Police and anti-fraud agencies were frequently mentioned by participants. This suggests that if such agencies maintained adequate connections with communities, they could work more effectively in investigating scams. According to the participants, victims may not know where or how to report incidents. Furthermore, they do not necessarily know what evidence needs to be turned in to ensure a successful investigation, which is another barrier to reporting.

> *"They need someone with the police or some agency to sit down with them and help decipher what has happened and give them the confidence to file charges."*

> *"Knowing they need to contact their bank, get a checklist, and what to look for in terms of online scam, and that it's okay to just delete an email without reading it."*

This is a step beyond merely getting notifications from the bank or the police about common scam scenarios. Participants suggested that prevention only works if it is provided personally.

### 4.2.2. Theme 5: "They Need to Be Told That This Happens a Lot"—Reassurance Is Encouraging

Participants asserted that victims need to be told about the nature of scams in order to be able to make informed decisions about reporting or asking for assistance. Perhaps because of the negative feelings associated with scam victimization, victims are not accessible through prevention and awareness-raising programs, especially when they deny their own victimization.

> *"They would need some kind of message to them that tells them that this happens a lot. That they are not alone. And here is how you report it."*

A related concern of the interviewees was that success stories are hardly covered by the media. If victims do not know there is a chance to recover lost funds, they will continue to assume that there is no use in filing reports.

> *"I would suggest something that says these people are caught. If that's true at all... [soft laughter] If you can get your money back, if that ever really happens, say that, because that increases sense of efficacy and taking the action or the effectiveness of the action."*

Participants acknowledged the existence of many prevention programs offered by agencies such as the AARP. Local police also provide some awareness-raising for residents, even in senior living facilities and nursing homes. However, participants vocalized critiques regarding such programs' suitability for older age groups, as well as the inaccessibility of materials and programs. Victims with emotional and psychological needs are especially reluctant to attend lectures provided by law enforcement.

> *"Admitting that you made a mistake is really hard sometimes. And if it's a mistake that has an impact, not just on you but your family, like a financial mistake, then that can be really, really scary and upsetting to people. That is why, normally we don't see any victims at local police meetings."*

### 4.3. Technical Needs
### Theme 6: "Together, They Went Online and Figured It Out"—Technical Help from Trusted Individuals

In most unreported cases, victims thought they should not bother with reporting because they had not lost a significant amount of money, or had inferred from the media that there was no chance to retrieve assets. However, even in these cases, victims still need assistance to ensure they never fall for scams again.

> *"These are folks that professionally never used computers until the very end and even then, all they did was word processing, that kind of thing; they just don't understand the dynamics."*

Scams are often not technical; instead, scammers use psychological manipulations to trick victims out of personal information or money (Atkins and Huang 2013). Nevertheless, technical help would be useful to check whether scammers left traceable information on the computer or whether they can still access valuable information. However, even in these situations, scam victims are often reluctant to ask for help, as they fear not being able to explain the situation. In addition, victims may lack computer-related linguistic knowledge, which makes them even more reluctant to talk to someone, especially when there are no trusted relatives or friends around. It is evident though, that even technical help should be personal, encouraging, and supportive. This brings the notion of "togetherness" into solving the problem and provides agency to the victim, who should be a part of investigating and resolving the case.

> *"By this time, my son-in-law had returned and so they began to ask questions and listened to me. So they listened to muster wherein gave me emotional support as a trusting person.*

*And then I have another son-in-law who is a computer science person. And together they began to say 'Okay, how can we make sure it won't happen again?'"*

*"Together, we went online and figured out where he [the scammer] had added an account to my computer so that he could probably come back check it later."*

*"Between all of that, and, of course, that helped me emotionally feel better about the fact that we may have control over it."*

Even when technical help was provided, the personal support of relatives and community members was an important component for victims to utilize in restoring self-efficacy and developing resiliency to scams. According to participant suggestions, said resiliency would be necessary for not only owning victimization in the first place but also for becoming able to identify red flags and judge situations that are too good to be true.

## 5. Discussion

### 5.1. Research Goals and Methods

Utilizing multiple non-random sampling methods, Virginia residents (*n* = 35), aged 60 and above, were asked to share their insights about why they think older people do not report scams to authorities and do not ask for help. By asking these questions, the researchers hoped to reveal personal narratives of scam victimization, including whether the help provided by family or community was useful. In exploring the needs of older victims (primary informants) and community members (secondary informants), this research revealed a staggering dearth of adequate responses. This research, applying the interpretive phenomenological analysis approach, provides insight into the unique personal experiences of the subjects concerning scam victimization as a major life event. While there is a growing body of research relying on quantitative data about online fraud (Li et al. 2022; Zhang and Ye 2022; Norris et al. 2019), qualitative empirical research about victims' feelings, needs, and overall concerns about scams is scarce (for exceptions, see Cross et al. 2016; Kopp et al. 2015). This project aimed to bridge this gap and propose practical recommendations.

### 5.2. Losing Trust: A Two-Way Mechanism

The current research finds that the rational decision-making process fails in highly manipulative scams. Out of the economic, neighborhood/social, and psychological factors that affect rational decision-making (Goudriaan et al. 2006), this research identified psychological factors as the ones most hindering rational decision-making in older scam victims.

The research applied multiple sampling techniques to recruit participants. The fact that snowball sampling was the most successful recruiting method reveals a lot about the hidden nature of scam victimization. After being victimized, people lose trust in communicating with strangers online or on the phone, which makes it difficult to access the target group. In accordance with this finding, other research has also associated being scammed with a loss of trust (Cross et al. 2016). This lack of trust that victims develop entangles every personal relationship around them. It affects victims' self-confidence (losing trust in oneself), their relationships with family members, relatives, neighbors, and law enforcement overall (losing trust in potential helpers). On the other hand, potential helpers also lose trust in the victim (i.e., in their ability to adequately cope with the situation, make good decisions about finances and spending, and avoid future scams).

In searching for reasons for underreporting, the current research also revealed that cognitive dissonance (for a summary, see Harmon-Jones and Harmon-Jones 2007) is fairly common among scam victims who, in denial of their victimization, may become more susceptible to being re-scammed. Cognitive dissonance (Festinger 1957) is based on the tenet that humans need their actions and beliefs to be consistent. When that is not the case, discomfort and tension arise. People naturally want to reduce the tension to return to harmony and justify their efforts. Scam victims achieve a sense of consistency and justify their efforts (e.g., in getting the promised fiscal reward in financial scams or emotional reward in romance scams) by convincing themselves that the act was not a scam, that they

had not lost money, or, despite all odds, that they are still going to get paid. That way, scam victims find themselves in intricate situations where they start to distrust family members and friends while instead trusting the scammer. Looking at the narratives that victims and secondary informants provided, it became apparent that scammers make perfect use of both the two-way mechanism of losing trust and cognitive dissonance in victims. The above mechanisms make victims reluctant to disclose victimization, on the one hand, and increase the odds of re-victimization on the other.

### 5.3. Older Victims of Scams Face More Negative Reactions than Younger Victims

According to research, most scam victims do not have dementia (James et al. 2014), although cognitive decline may apply (DeLiema et al. 2020; Shao et al. 2019). Their victimization may be rooted in newly acquired digital skills (Mitzner et al. 2019), loneliness and isolation (Alves and Wilson 2008), a low level of self-control (Parti 2022; Ngo and Paternoster 2011), or over-confidence stemming from a higher-than-average level of education and income/wealth (Vandenbroucke and Zhu 2017). Despite these risk factors, it is important to note that being tricked into paying for something without getting anything in return is not a situation unique to older people. However, older victims tend to receive more negative reactions, anger, and disappointment, and harsher preventative measures than younger victims of scams. Older people can lose trust in their loved ones or entire family relationships can be broken. Victims can be forced under family members' supervision and become painfully dependent, both financially and physically. While we agree there are situations in which radical responses are necessary, all victims deserve to be listened to, and their loss and feelings deserve understanding and acknowledgment. Thus, active listening techniques are recommended in communicating with victims of crime in general (Gonzales et al. 2005). In addition, trauma-informed communications (Stubbe 2013) should be applied by families, friends, and community members in order to decipher the concerns and needs of older scam victims.

### 5.4. Building Resilience by Providing Technical and Emotional Help through Prosocial Connections

Given the highly manipulative nature of scams, technical skills might be useful but not essential in prevention. Instead, recognizing the signs of social engineering-based manipulations applied by scammers (Atkins and Huang 2013) would be essential. However, technical knowledge can still make it possible to identify traces of scams on victims' computers. The current research finds that asking for and accepting technical aid is more likely when such help comes from trusted individuals (relatives or close family members), instead of from IT professionals or law enforcement who are strangers to the victim. Close proxy, trusted individuals might even strengthen critical interpersonal relationships, which can help protect against re-victimization. By building resiliency and restoring a sense of agency, trust, and self-confidence in victims, technical help would complement the emotional and psychological support which loved ones should provide.

### 5.5. Multi-Generational Awareness-Raising and Prevention Programs Needed

The current research identified the need to be able to turn to trusted community members who, after carefully listening to victims, could provide a second opinion on scam-related situations. Trusted community members would not only serve as additional eyes and ears but would also be the only imaginable way for some people to discuss red flags without losing face out of guilt and self-blame. By developing community support, victims could viably be provided immediate feedback and situational assessment by consulting with independent individuals or experts.

The need for community intervention boils down to expanding awareness-raising activities and programs for multiple generations. Intergenerational buddy systems where members from different generations pair up to coach each other in building online technical and social skills can help achieve an organic, community-based support system such as Sunshine Connected (Center for Aging and Brain Health Innovation n.d.) or the Silverling

Buddy System (Jewish Boston Teens 2020) against social isolation and stigmatization. Such programs have demonstrated success in dementia patients (Gerritzen et al. 2020; Houghton et al. 2022) and aged care facilities (De Bellis et al. 2022), just to name a few. Community-based programs can help maintain prosocial connections and provide greater consideration in situations involving suspicious calls or emails.

*5.6. Strengths and Limitations*

A few limitations should be considered while reading the results of this research. First, the study only represents victims' needs residing in the Commonwealth of Virginia, without focusing on the demographics of race, gender, or other social characteristics apart from their age being above 60 and having been targeted by scams. Since all the volunteered participants had been interviewed and were not selected based on demographic indicators, the sample cannot represent a diverse population. Most respondents were white Caucasian, cis-gender females; therefore, future selection of a larger sample with more diverse demographics might provide different results.

Furthermore, more than half of the participants were secondary informants from the victim's family or close friends. This–on top of the emotional and psychological impact of scams discussed above–suggests that it is very difficult to find scam victims willing to share their experiences. Partly because of the barriers to getting in touch with primary victims, the sampling was purposeful and relatively small, so the findings cannot be generalized.

The researchers recognize that subjective judgment is not alien to research. Hence, the power dynamics between the storyteller and listener are essential to acknowledge when dealing with an interpretive paradigm. Analyzing what participants say depends on the researcher's paradigmatic axioms, belief system, and background information (Bailey 2007, pp. 159–73). This research acknowledges that, though participants had full freedom to express themselves during the interviews, the conceptual semantic interpretation was made by the researcher. Since participants were not involved in the transcribing and interpreting stage, the final text is the researchers' understanding of the phenomenon under study.

This research also acknowledges the environmental factors of the interviews, which took place amid the COVID-19 pandemic. Broom et al. (2009) looked at how contextual, biographical, and psychological variables affected the flow and content of interviews. The narrative may also be influenced by the outer environment for both researcher and interviewee as the long-term lockdown and social isolation may have had a varied effect on the psychological well-being of individuals; hence, the same research in different environments might have different effects.

Although it cannot be generalized, the findings corroborate previous research findings emphasizing the role of trust in authorities, as well as the role of family and community support in encouraging reporting. It is well known from victimology (e.g., Lonsway and Archambault 2021; Maddox et al. 2011; Patterson 2011) that victims establish a better rapport with police if their concerns are listened to and their listeners show empathy towards them. If law enforcement and the criminal justice system can help victims build trust in them, subsequent offenses are more likely to be reported, researchers and policymakers will know more about victimization, and hence, better prevention and intervention programs can be developed.

In addition, this research adds to the very small number of projects that examine the scam victimization of older individuals. DeLiema et al. (2020) find that individuals may be more comfortable speaking about their experiences with someone in person who can establish the credibility of the organization and build rapport versus over the phone, where the caller's identity and intentions are harder to verify. By reaching out to victims and their communities directly, the current research fills this gap by providing invaluable insights into the personal narratives of victims, which could never be revealed by studies solely reliant on quantitative methods.

While it is necessary to equip police with the right tools for tackling scams, it is also necessary to employ preventative measures such as community education for internet

literacy and security precautions, as well as by creating emergency resources for contacting if any suspicious activity is detected. Furthermore, it is essential to start removing the taboo around discussing online scams by providing older people with resources and emotional support, and by not blaming them for their victimization. Future research should investigate this issue, specifying questions about family and community support for victims of online scams.

## 6. Conclusions

When it comes to asking for help or reporting an incident, a victim's decision-making process is not entirely rational. In contrast, the current study finds that the emotional and psychological impact of scams is often more tolling than the financial impact. Hence, shame, guilt, self-blame, and fear of losing control over assets are significant drivers in keeping one's victimization secret. The same difficult feelings, paired with cognitive dissonance, can alienate victims from loved ones and the community, thus compounding a sense of isolation and increasing the odds of future re-victimization. Researchers identify isolation as a form of psychological manipulation used in offenses (e.g., in intimate partner violence: Pence and Paymar 1993; Stark 2007) or financial abuse by relatives and caretakers (DeLiema 2018). The current research posits that social isolation is a technique also employed by highly manipulative fraudsters (scammers), which makes asking for help and reporting more difficult. Thus, future prevention efforts must unravel the "trust issue" by examining what roles different agents (family members, friends, neighbors, caretakers, law enforcement, and anti-fraud agencies) could have in rebuilding interpersonal trust. This could help older people develop stronger resilience against scammers.

**Author Contributions:** Conceptualization, K.P.; Methodology, F.T.; Software, K.P.; Validation, F.T.; Formal Analysis, K.P.; Investigation, K.P. and F.T.; Resources, K.P.; Data Curation, K.P.; Writing–Original Draft Preparation, K.P. and F.T.; Writing–Review and Editing, K.P.; Visualization, K.P.; Supervision, K.P.; Project Administration, F.T.; Funding Acquisition, K.P. All authors have read and agreed to the published version of the manuscript.

**Funding:** The research was funded by the Institute for Creativity, Arts & Technology, the Center for Gerontology, and the Center for Peace Studies and Violence Prevention at Virginia Tech. This publication was processed with the support of the Niles Research Grant at Virginia Tech, and Virginia Tech's Open Access Subvention Fund.

**Institutional Review Board Statement:** The study was conducted according to the guidelines of the Belmont Report and the Declaration of Helsinki, and approved by the Institutional Review Board of Virginia Tech (protocol code #21-415, 15 June 2021).

**Informed Consent Statement:** Informed consent was obtained from all subjects involved in the study.

**Data Availability Statement:** The data presented in this study are available on request from the corresponding author.

**Acknowledgments:** The authors like to thank Pamela B. Teaster Professor of Gerontology who participated in the coding process. We also thank Virginia Tech research assistants Elizabeth Harrell, Patrick McElligott, and Zoe McCray.

**Conflicts of Interest:** The authors declare no conflict of interest.

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
