# Peer review of "“If We Don’t Listen to Them, We Make Them Lose More than Money:” Exploring Reasons for Underreporting and the Needs of Older Scam Victims"

_socsci, doi:10.3390/socsci12050264_

Round 1
Reviewer 1 Report
Thank you for the opportunity to review this manuscript which touches on the critically important topic of addressing scams among older adults. This reviewer found the manuscript to be well-organized, cohesive, and engaging to read. The following minor edits will improve this manuscript and assertions contained therein:
Lines 41-50: This large chunk of text in the literature review section needs proper citation
Methods section around line 237: How did you reach potential interviewees who were hesitant to use phone and internet? Was it a warm hand-off? Did they ask individuals to pass on contact information to the potential interviewee?
Methods, around line 246, recommend adding additional demographic data. These data are discussed within the limitations section but not fully described here, including race/ethnicity, # of primary and secondary informants, age ranges, etc.
Results: To frame up the findings, add a paragraph or two at the beginning of the results section to briefly describe the range of scam experiences that were discussed, if these are known, including financial impact (in dollars) and type of scam. For example, did respondents discuss romantic scams, gift card scams, etc.
Tagging the individual excerpts in the results section by primary and secondary informants and other identifying characteristics would help to contextualize the quotes.
Did secondary informants mostly discuss a given theme as compared to primary informants? Add these types of observations to each thematic section if there are comparisons to be drawn. Such findings have implications for outreach and education to this target population vs. secondary informants/helpers.
Line 456: The teme 5 subheader is bolded when others are not bolded elsewhere in this section.
Author Response
Please see the attachment. Thank you for your work as a reviewer.

Reviewer 2 Report
Dear authors, I present my general considerations about the manuscript, which I found very interesting:
Line 49: The abbreviation "IRS" is exposed without a meaning. Perhaps it is understandable for those who live in the US, but I believe it would be interesting for other readers.
Lines 228 and 229: I understand this as an anticipation of the results obtained.
Lines 252 and 253: Has the project not been submitted to a research ethics committee?
Line 475: The abbreviation "AARP" is exposed without a meaning, an argument that is based on the same concerns raised about line 49.
Line 645: Regarding the exposed limits, which I consider necessary and valid, there is no information, at least I did not observe, that half of the participants were secondary informants.
Author Response
Please see the attachment. Thank you for your work reviewing our manuscript.
